# Acid Soil Improvement Enhances Disease Tolerance in Citrus Infected by *Candidatus* Liberibacter asiaticus

**DOI:** 10.3390/ijms21103614

**Published:** 2020-05-20

**Authors:** Bo Li, Shuangchao Wang, Yi Zhang, Dewen Qiu

**Affiliations:** 1The State Key Laboratory of Plant Diseases and Insect Pests, Institute of Plant Protection, Chinese Academy of Agricultural Sciences, Beijing 100193, China; bo.Li@student.uliege.be (B.L.); wangshuangchao@caas.cn (S.W.); yizhang@bjfu.edu.cn (Y.Z.); 2Functional and Evolutionary Entomology, Gembloux Agro-Bio Tech, University of Liège, B-5030 Gembloux, Belgium

**Keywords:** Huanglongbing, plant immune response, soil acidification, salicylic acid, root metabolic activity

## Abstract

Huanglongbing (HLB) is a devastating citrus disease that has caused massive economic losses to the citrus industry worldwide. The disease is endemic in most citrus-producing areas of southern China, especially in the sweet orange orchards where soil acidification has intensified. In this work, we used lime as soil pH amendment to optimize soil pH and enhance the endurance capacity of citrus against *Candidatus* Liberibacter asiaticus (CLas). The results showed that regulation of soil acidity is effective to reduce the occurrence of new infections and mitigate disease severity in the presence of HLB disease. We also studied the associated molecular mechanism and found that acid soil improvement can (i) increase the root metabolic activity and up-regulate the expression of ion transporter-related genes in HLB-infected roots, (ii) alleviate the physiological disorders of sieve tube blockage of HLB-infected leaves, (iii) strengthen the citrus immune response by increasing the expression of genes involved in SAR and activating the salicylic acid signal pathway, (iv) up-regulate 55 proteins related to stress/defence response and secondary metabolism. This study contributes to a better understanding of the correlation between environment factors and HLB disease outbreaks and also suggests that acid soil improvement is of potential value for the management of HLB disease in southern China.

## 1. Introduction

Huanglongbing (HLB) is one of most destructive citrus diseases worldwide, which is caused by the phloem-restricted bacterium *Candidatus* Liberibacter asiaticus (CLas) and is easily transmitted through the citrus psyllid (ACP) *Diaphorina citri* [1]. CLas infection usually induces yellow shoots, blotchy mottled leaves, root degradation and decay, and ultimately tree death [2]. HLB disease has the characteristics of rapid spread and is extremely difficult to be eradicated. Once symptomatic citrus trees sporadically occur in a citrus orchard, the disease will spread throughout the whole orchard within two to three years, causing the citrus trees to lose their productive capacity [3]. In China, the disease has been reported for more than two centuries and can affect all commercial citrus varieties [4]. Currently, the disease has spread to more than 11 citrus-growing regions especially in Guangdong, Hunan, Guangxi and Jiangxi provinces [5]. Since 2012, just in Ganzhou, Jiangxi province, more than 45 million infected trees have been removed, and the direct economic loss is estimated to be more than 9 billion yuan [6]. Therefore, there is an urgent need to develop a sustainable solution to suppress the spread of HLB disease and enable the continued economic production of CLas-infected citrus trees.

The traditional control measures of HLB disease include spraying insecticides for vector control, directly injecting antibiotics into the trunks of citrus trees, thermotherapy and use of nursery stock free of CLas etc. [7,8]. However, excessive pesticides can cause environmental pollution, insecticides and antibiotics left on the citrus are harmful to human health and thermotherapy has high requirements on the heating device and orchard terrain, and it also requires a lot of labour costs. Therefore, it is imperative to develop a novel and economic treatment strategy of HLB. It is well-known that the CLas bacterium, psyllid vectors, citrus hosts and the environment constitute the HLB disease pyramid [9], and each factor is necessary for development and prevalence of HLB disease. Previous studies have mainly concentrated on controlling citrus HLB disease by eliminating CLas bacterium and psyllid vectors and rogueing HLB-infected trees; there are few studies on the role of citrus growing environment, especially soil conditions on the management of HLB disease. 

Healthy soil is one of the important environmental factors that affects plant growth and stress resistance, because it not only provides the essential water and nutrients for plants, but also guarantees the establishment of a well-functioning soil ecosystem [10,11]. However, in the past 20 years, the soil pH of arable land in southern China has decreased significantly. Soil acidification is mainly due to the excessive application of chemical fertilizers, especially nitrogen fertilizers [12]. Many researchers reported that soil health is associated with limited disease outbreaks, and indeed, many plant diseases were more serious in acidified soils compared with healthy soils [13,14]. Soil acidification leads to increased toxicities of metal cations (i.e., Al^3+^, Mn^2+^) to plants and accelerates phosphorus and base cation (i.e., Ca^2+^, Mg^2+^, K^+^) erosion, which has negative effects on plant disease progress and nutrition absorption [15,16]. Another consequence of soil acidification is destruction of the microecological balance and microbial community diversity, which leads to the loss of a stable and healthy soil environment and weakens the resistance of plants to pathogenic bacteria in acidic soil [17]. Although extensive research has shown that soil acidification positively affects the development of various plant diseases, there has been no detailed investigation of the correlation between the occurrence of HLB disease and soil pH values.

At present, there are still large numbers of citrus orchards with HLB disease in southern China, these HLB-endemic regions are at great risk for soil acidification and HLB spread. Due to the lack of effective control measures, infected citrus trees have a noticeably shortened profitable life and dramatically declined yield, and they have to be removed to prevent the spread of the disease [18]. Therefore, on the premise of controlling the spread of HLB disease in highly endemic citrus-producing regions, the critical challenge in southern China is to restore and maintain the health and productivity of citrus trees in the presence of soil acidification and HLB disease. In our study, we added lime as a soil pH ameliorant in citrus orchards with acidic soils to slow the disease progress and reduce the spread of HLB disease. At the same time, the mechanisms of acid soil improvement on management of HLB disease were analysed by comparing the changes of citrus root tissue vigour and plant immune related resistance genes and identifying the differentially expressed proteins in midrib tissues before and after soil treatment. This study will provide a novel perspective on managing HLB disease using an eco-friendly approach.

## 2. Results

### 2.1. Effect of Soil Ameliorators on pH Value of Acidic Soil in Citrus Orchard

A soil pH survey of the experimental field showed that there was a serious degree of soil acidification. Although soil pH values fluctuate slightly, most of the orchard measurement points selected in this experiment had pH values between 4.5 and 5.5 (Figure 1A). In the present study, the pH of the soil in the citrus orchard was restored to 6.0–6.5 in five months after treatment by adding appropriate lime (Figure 1A), which is suitable for growth of citrus plants. Subsequent investigation results showed that the application of soil amendments in our study could maintain the stability of soil pH value for at least one year (Appendix A).

### 2.2. Effect of Acid Soil Improvement on Management of HLB Disease in Southern China

#### 2.2.1. HLB Infections Rate

The new infections rate of HLB disease in the FCK (control grove in the field trial) and FT (treatment grove in field trial) was investigated using PCR assay at the fifth month after application of soil ameliorants in the field. The results showed that the average natural incidence rate of CLas was 20.2% in the FT and 32.2% in the FCK (Figure 1B). In the following year, we continued to monitor the infection rate in both groves. The rate of HLB infections in FT was about 61%, and most of the HLB-infected citrus trees presented mildly infected symptoms, whereas more than 95% of plants in FCK were infected, and most of the citrus trees presented seriously infected symptoms (Appendix A). At the same time, the proportion of CLas bacterium in new shoots of the HLB-infected citrus was determined. The proportion of detectable CLas pathogens in the FCK was over 93.0%, while it was about 60.0% in the FT (Figure 1C). These results indicate that regulation of soil acidification is beneficial for limiting the movement of CLas bacterium to new shoots in infected plants and inhibiting the spread of HLB disease to healthy plants.

#### 2.2.2. Chlorophyll Content and Disease Index

We also evaluated the HLB disease index by calculating the yellowing rate of the canopy upon different soil pH conditions in the field. In the FT, the disease index of about 70% of HLB-infected citrus trees was level 1 or level 2. While in the FCK, the disease index of 80% of citrus trees was mostly estimated to be at level 3, level 4 or level 5 (Figure 2C). In the case of the greenhouse trial, many of the citrus leaves in the GCK (control group in greenhouse trial) showed obvious symptoms of spotted-yellowing at the fifth month after soil treatment, whereas typical symptoms did not appear on the leaves of GT (treatment group in greenhouse trial) (Figure 4A). Meanwhile, the relative chlorophyll content of the leaves was measured. The chlorophyll content in the FT was significantly increased (about 10%) than that in the FCK (Figure 2A). The result of the greenhouse trial showed that the chlorophyll content of citrus leaf from soil with pH 6.5 conditions was also 15% higher than that from pH 4.5 soil (Figure 2B). The infected leaves in the GCK not only showed more obvious symptoms in appearance, but it was also found that the plugging of their sieve tubes was more exacerbated than that in the GT, using SEM observation (Figure 4). It was speculated that HLB-infected citrus plants grown in suitable soil pH conditions were more likely to delay disease progression than those grown in acidic soils.

#### 2.2.3. The titre of CLas bacterium

The titre of CLas in the leaf midribs were quantified monthly using q-PCR after treatment in the orchard. The titre of CLas showed a continuous downward trend in both groves, but the decreasing trend was more obvious in FT. Especially at the fifth month, the titre of CLas was about 3 times lower than that in the FCK (Figure 3A). In the greenhouse trial, the titre of CLas increased slightly over time under acidic soil conditions (pH = 4.5), and it was about 3.5 folds decreased under amended soil conditions (pH = 6.5) compared with the control sample at the fifth month (Figure 3B). The above results suggest that regulation of soil acidification effectively inhibited the accumulation of the CLas in the infected plants.

#### 2.2.4. Fruit Yield and Quality

Soil acidification accelerated the loss of production capacity and reduced the fruit quality of HLB-infected citrus trees. The average yield of the infected plants in the FT was 57.6 kg per tree, which was higher than in the FCK (52.0 kg per tree) at the first year after soil treatment by liming (Table 1). The fruit yield survey in the following year found that the average yield of HLB-positive plants in the FT remained above 50 kg per tree, however, in the FCK, the yield of HLB-positive plants dropped sharply, with an average yield of 27.1 kg per tree, some of which lost their economic value because the yield was below 20 kg per tree (Appendix A). Meanwhile, the diameter of the fruit in the FT increased by 8.42%, and the fruit drop rate markedly reduced by 50% compared with that in the FCK (Table 1). It was observed that the CLas-infected fruits in the FCK were small, poorly coloured, misshapen, whereas the infected fruits in the FT presented more like healthy fruit in visual appearance (Appendix A). The results of assays of fruit quality indexes showed that in the FT, three monosaccharides, such as glucose (2.5 g × 100 g^−1^), sucrose (4.4 g × 100 g^−1^), fructose (2.4 g × 100 g^−1^), and the soluble solids content (>12%) of fruits (Table 1) basically reached the nutritional requirements of edible fruits for commercial use, while the diseased fruits in the FCK failed to meet the requirements according to the Green food-Citrus NY/T 426-2000 of Agricultural Standards of China. Based on the above results, it is inferred that providing suitable growth soil conditions for citrus trees infected with HLB disease is more likely to produce asymptomatic infected fruits and maintain the yield and quality of infected plants.

### 2.3. Effect of Soil pH on the Expression of the Ion Transport-Related Genes and Metabolic Activity in the HLB-Infected Roots

In the greenhouse trial, the activity of CLas-infected roots grown in the GT was determined using the TTC (2,3,5-triphenyltetrazolium chloride) reduction method, which was significantly enhanced compared with that in the GCK (Figure 5A). The roots in the FT also possessed higher root viability than those in the FCK in the field trial (Appendix A). The lateral roots of seedlings in the GCK showed sparse, short and stunted symptoms compared to those in the GT (Figure 4B). Furthermore, the expression level of four genes, *CsAmtB* (ammonium transporter), *CsVIT1* (vacuolar iron transporter), *CsZIP1* (zinc transporter) and *CsBOR1* (boron transporter), associated with ion transporters in greenhouses were measured using q-PCR. In the GCK, the expression level of the four ion-transporting genes in root tissues of CLas-infected seedlings was respectively down-regulated by 4.76, 1.92, 1.63 and 2.85 times compared to that in the GT (Figure 5B). These results indicate that the regulation of soil acidification had a beneficial influence on the root metabolic activity and the nutrient uptake ability of the HLB-infected seedlings.

### 2.4. Effect of Soil pH on the SA Content and Expression of the Resistance-Related Genes in the HLB-Infected Plants

In this study, the content of endogenous hormone SA and the expression of genes involved in host disease resistance were determined in the greenhouse trial. The results showed that the SA content in the roots and midribs in the GCK was 1.55 and 1.67 times lower than in the GT (Figure 6B,D). Additionally, in the leaves and roots, the expression of resistance-related genes in the GT was up-regulated significantly more than that in the GCK. q-PCR analysis indicated that in infected roots, the pathogenesis-related proteins genes, *CsPR1* and *CsPR5*, were meaningfully higher in the GT than that in the GCK. In addition, the *CsNPR1* and *CsALD1* genes were involved in plant systemic acquired resistance (SAR) and were up-regulated 4.75-fold and 5.78-fold, respectively. Meanwhile, in leaf midribs, the expression levels of the *CsPR1*, *CsPR5*, *CsNPR1* and *CsALD1* genes of HLB-infected seedlings in the GT were up-regulated more than those in the GCK (Figure 6C). The expression levels of the other two WRKY transcription factors in leaves and roots were both remarkably increased (Figure 6A), which is related to the resistance in the biotic and abiotic stress pathway [19]. The above results indicated that soil improvement had an affirmative effect on the expression of the resistance genes and the SA signalling hormone synthesis in HLB-infected citrus plants.

### 2.5. Proteome Characteristics of Midribs of HLB-Infected Plants Grown in Different Soil pH Conditions

TMT-labelled proteome analyses were applied to understand the metabolic process of midribs of HLB-infected citrus under differing soil pH conditions. We identified 119 differentially expressed proteins at a cutoff value of >|±1.5|-fold (*p* value < 0.005) (Appendix A), including 55 up-regulated and 64 down-regulated proteins, in the midribs sample in liming-amended soils (pH = 6.5) compared with the midribs sample collected from untreated soils (pH = 4.5). Most significantly, most of the up-regulated proteins are involved in stress/defence responses and secondary metabolism. The stress/defence response-related proteins mainly include two pathogenesis-related proteins, five methyltransferases and a probable leucine-rich repeat receptors and the secondary metabolites biosynthesis pathway-related proteins, including three dirigent proteins, three serine-type carboxypeptidases, a terpene synthase, a 3-beta-hydroxy-delta5-steroid dehydrogenase and a UDP-glucosyltransferase (Table 2). Most down-regulated proteins were annotated as unknown function or heat-shock proteins that can protect other proteins against heat-induced denaturation and aggregation. The details of the differentially expressed proteins are listed in Appendix A. Based on analysis of comparative proteomics data, it is speculated that defence response of infected plants may be more intensely activated in liming-amended soil than in heavily acidic soil.

## 3. Discussion

The health of soil determines agricultural sustainability and environmental quality and greatly affects plant vigor, productivity and disease resistance [20]. Previous reports have showed that the occurrence of multiple plant diseases (fungal, bacterial, viral and oomycete) is closely related with poor soil conditions and is particularly influenced by both increases (liming) and decreases (not liming) in soil pH [21,22]. It has been reported that soil acidification can cause the outbreak of bacterial wilt disease in southern China, and researchers have reduced the incidence of the disease by improving soil pH [14]. Rapid soil acidification has become a serious global problem and is one of the manifestations of soil degradation. In southern China, there is a general trend of increasing soil acidification in the HLB-endemic citrus-producing regions. For example, the soil pH of over 1000 samples from 477 orchards in 18 counties of Ganzhou, Jiangxi Province, where we conducted the field experiment, were seriously acidic, and the average soil pH value was 4.6 [23]. Heavy soil acidification also exists in other major citrus growing areas such as Fujian, Guangdong, Guangxi, Hubei, Hunan and Zhejiang provinces, where HLB disease is highly endemic [24,25,26,27,28]. In contrast, there is no large-scale outbreak of HLB disease and soil acidification in the citrus growing area of Chongqing, where more than 80% of citrus orchards have a soil pH of approximately 6.5 [29,30,31]. Our study indicated that soil acidification occurred seriously in our experimental orchards, and it accelerated the spread of HLB disease in citrus orchards and promoted movement of CLas to new shoots of infected plants (Figure 1B,C). These results indicated that soil acidification is related to the occurrence and movement of HLB disease. Further experiments showed that when the acidic soil was ameliorated, the cumulative amount of CLas in citrus leaves was significantly lower than that of the untreated control group, and the severity of HLB disease was relatively mild (Figure 2C and Figure 3). According to these data, we can infer that soil acidification promotes the occurrence of HLB disease, but molecular understanding of the process remains unclear, and acid soil improvement is worth considering as a part of management strategy to control HLB disease.

The application of soil pH amendments is an effective way to improve acidic soils. Liming is a well-established soil pH remediation agent used to optimize soil pH for agricultural production [22]. It can enhance nutrient cycling of the soil environment and decrease uptake of toxic heavy metals to benefit yields [32], thereby reducing the impacts of soil acidification on the growth vigour of citrus plants. In this study, we found that the average pH of the orchard soil in the FT increased by 1–1.5 units and maintained a stable status for a long period after reasonable application of lime (Figure 1A and Appendix A). Meanwhile, our research results showed that the yield of infected plants was also improved by regulating the acidic soils. These results indicate that lime is an ideal agent to alleviate soil acidification in citrus orchards and is beneficial to the quality and yield of citrus in acidified soil.

Phloem-plugging and yellow shoots are the typical symptoms of HLB disease. Previous studies found that the sieve pore plugging of phloem was normally caused by ultrastructural deformation of phloem cells and accumulation of starch in the sieve elements [2]. Concurrently, the passive feedback of excessive starch accumulation leads to the inhibition of photosynthesis in infected leaves [33], which might be one of the reasons for the mottled yellowing of infected leaves. Our results have demonstrated that the HLB-associated symptoms and disease severity of infected citrus plants were alleviated in liming-amended soils. The content of chlorophyll was greatly increased (Figure 2A,B) and the plugging of sieve pores was moderately mitigated in CLas positive leaves under amended soil condition (Figure 4B). The proportion of asymptomatic branches in the canopy of HLB-positive citrus plants grown in the FT is significantly higher than that in the FCK (Figure 2C). These changes would be of advantage to promote photosynthesis and transportation of photosynthate in citrus plants and provide a stable and favourable physiological basis for the maintenance of yield. Taking the above results into account, we may reach the conclusion that although acid soil improvement cannot completely control HLB disease, it has the potential to delay the progression of HLB disease by applying lime to regulate soil acidification.

Preserving the health of roots is a key component in HLB disease management because plant root systems perform many essential functions. Previous study has proved that plants grown in low pH soil have weaker plant growth and are more susceptible to phytopathogens [20]. One main reason is that soil acidification immensely alters physical and chemical properties and decreases soil quality. Soil acidification can cause a deficiency of essential nutrients and decrease the cation exchange capacity, accompanied by increased toxicities of toxic metals [16]. In acidic soils, the availability of soil N, P, Mg and K drops dramatically, thus reducing the efficiency with which plants make use of fertilizers. Another reason is that long-term growing in acidic soils directly impairs the micro- and macronutrients uptake ability of plant roots [34]. As the soil pH drops to below 6.0, there is a deleterious effect on the morphological and physiological development of plant roots with the Al^3+^ concentration increasing, which subsequently antagonizes water and essential nutrients uptake in the plant [35]. In the case of citrus plants heavily infected with HLB pathogens, the growth and development of the roots are also greatly inhibited [4]. CLas infection disrupts the transportation of photosynthate to root tissue [36] and eventually causes substantial root dieback due to carbohydrate starvation and cationic toxicity in acid soils. In the greenhouse trial, the results showed that the roots of infected citrus in liming-amended soils (pH = 6.5) had better morphological development and significantly improved metabolic activity (Figure 4A and Figure 5A), and the expression of some genes related to nutrient uptake were up-regulated compared with that in acidic soil (pH = 4.5) conditions (Figure 5B). These results suggest that raising the pH value of acidified soil can invigorate the absorption of nutrients and maintain the root health of HLB-infected citrus trees, thereby indirectly enhancing the endurance of citrus to HLB disease.

Plant innate immunity is the front line of defence against pathogen invasion [37]. CLas infections can cause nonself-recognition immune responses in citrus plants, including increased levels of salicylic acid and induction of the expression of immune response genes encoding PR genes, SA biosynthesis or degradation modulators and WRKY transcription factor, and so on [1]. The small phenolic compound SA plays an central regulatory role in immune responses that confer plant resistances to a variety of pathogens and SA-mediated immune responses essential for the activation of SAR [38]. Much research also showed that accumulation of SA in citrus could potentially lead to enhanced resistance to a range of pathogens [39,40]. In our study, results showed that the SA content of leaf midribs and roots in the GT was relatively higher than that in the GCK (Figure 6B,D), and the expression of genes *CsNPR1* and *CsALD1* were significantly up-regulated (Figure 6A,C). The most well-established role of *NPR1* and *ALD1* genes are as positive regulators of SAR and SA accumulation [38]. The expression of two *WRKY* genes, which are involved in the regulation of SAR, also were significantly increased in infected plants under liming-amended soils condition.

Studies in *Arabidopsis* have shown that the expression of pathogenesis-related genes PR-1 and PR-5 considered to be markers for SA-dependent SAR was highly induced in both roots and leaves of infected microorganisms [41]. In accordance with the previous report, our studies have demonstrated that the expression of pathogenesis-related genes *CsPR1* and *CsPR5* at the transcriptional and translational level were higher in the roots and leaves of HLB-infected citrus in liming-amended soils. The results indicate that regulation of soil acidification can indeed arouse the infected citrus in amended soils to maintain stronger immune responses for resisting the infection of the CLas bacterium. This is consistent with previous research that acid soil improvement increased the resistance gene expression and reduced the occurrence of bacterial wilt [14]. Therefore, we could tentatively attribute the observed defence regulation in our study to a “strengthened potential tolerance”, which improves SA-mediated immune responses against CLas by enhancing the citrus vitality and health.

Analysis of comparative proteomics data also indicated that the expression of proteins related to the synthesis of secondary metabolites, which contain antimicrobial properties or are associated with plant defence responses, were up-regulated in midribs of CLas-infected trees in liming-amended soils (Table 2), such as the dirigent-like protein family, methyltransferase family (MTase), terpene synthase (TPS), serine carboxypeptidase (SCP) and UDP-glucosyltransferase (UGT). The dirigent-like protein family contains a number of proteins involved in lignification which are induced during disease response in plants [42]. In particular, the up-regulated methyltransferase family proteins A0A067FUD3 and A0A067DZ96 are involved in the biosynthesis of methylsalicylate (MeSA) in response to stresses. The volatile compound MeSA is hypothesized to act as an airborne signal that triggers defence responses in uninfected plants [43]. TPS, SCP and UGT are also closely related to plant resistance. For example, TPS is a crucial enzyme in the synthesis of terpene plant metabolites, which act as defensive compounds against pathogens and herbivores [44]. The overexpression of the TPS gene, OsTPS19, in rice plants enhanced resistance against plant pathogen, while OsTPS19 RNAi lines were more susceptible to the pathogenic bacterium [45]. SCP-like protein SCPL1 (SAD7) in oat is required for the synthesis of antimicrobial compounds and for disease resistance [46]. UGT is involved in Fusarium head blight resistance in wheat [47]. Generally, when citrus was infected by the HLB pathogen, a series of metabolic changes and resistance responses were stimulated to resist the infection [2,48]. However, soil acidification may weaken the immune defence capabilities of citrus and inhibit the synthesis of antibacterial secondary metabolites. Combined with the findings of this study, we can conclude that soil improvement in acidified orchards contributes to exerting optimal resistance of infected citrus against HLB pathogen and their ability to sustain plant growth after infection.

The diversity and stability of the bacterial community present in the rhizosphere heavily influences plant growth and immunity [49]. Recent studies have demonstrated that soil pH was a predominant factor mediating the soil microbial community [50]. CLas infection was accompanied by a notable increase in the proportion of *Acidobacteria* in citrus rhizosphere [51]. According to our simultaneous analysis of bacterial communities in rhizosphere soil of HLB-infected citrus (unpublished data), the relative abundance of *Acidobacteria* on the phylum level was significantly reduced in cloned libraries from lime-amended soils (16.56%) compared with cloned libraries from acidic soils (23.71%) (Appendix A). The present study raises the conjecture that soil acidification probably provides a favourable external environment for citrus pathogenic microbes, and altering soil pH by liming will profoundly reshape the rhizosphere microbiome structure. Further investigations are needed to reveal the role of soil environment for native and beneficial microbiota in protecting citrus from CLas and other microbial pathogens.

## 4. Materials and methods

### 4.1. Plant Materials and Soil Treatment in Field Trial

The experimental orchards were located in the southeast of Ganzhou, Jiangxi Province, the primary citrus production area in China (latitude 25°83′N and longitude 114°93′W). The soil type in this area belongs to the typical red soils, and the citrus orchard has had a general tendency of aggravating acidification in recent years. The citrus trees are aged 8–10 years. The cultivar is sweet orange (*Citrus sinensis* Osbeck cv. *Newhall*) grafted onto citrange rootstock (*Citrus sinensis* [L.] Osb × *Poncirus trifoliate* [L.] Raf.). The orchard containing more than eight hundred trees was equally divided into two blocks according to the vertical direction of the row-planting pattern, which were named control grove (FCK) and treatment grove (FT). The average natural incidence of citrus tree HLB in the whole orchard is about 5%, according to the results of an epidemiological survey in December 2016. Three hundred trees were respectively selected from each grove in a completely randomized method to detect the incidence of HLB. Twenty HLB-positive citrus trees with similar disease severity (initial symptoms observed in the canopy is less than 20%) were selected from each grove for subsequent experiments.

To improve the soil pH conditions, lime was selected as the soil amendment to regulate the acidified soil. In the FT, lime (3000 kg × ha^−1^) purchased from a local factory was twice applied around the root crown by ploughing in April and May 2017, with an interval of 40 days per treatment. There was no treatment except ploughing in the FCK. For the duration of the experiment, unified irrigation and fertilization were carried in both groves.

### 4.2. Plant Materials and Soil Treatment in Greenhouse Trial

The experimental plants were 5-month-old sweet oranges (Citrus sinensis Osbeck cv. Newhall) on citrange (Citrus sinensis [L.] Osb × Poncirus trifoliate [L.] Raf.) rootstock. They were cultivated in 20 cm diameter × 30 cm deep pots with pH = 4.5 soil taken from the field. All experimental plants had been previously inoculated with CLas bacterium via ACP and confirmed HLB-positive using conventional and quantitative PCR. After two months of inoculation, eighty citrus seedlings were repotted in pH = 4.5 acidic soils and kept as control group (GCK); eighty citrus seedlings were repotted in pH = 6.5 liming-amended soils and kept as treatment group (GT). All plants were maintained under greenhouse conditions with the temperatures not exceeding 30 °C and supplemented with 16 h light and 8 h dark. The plants were watered as needed and fertilized once using a commercial slow-release fertilizer with NPK ratios of 16:8:18 at 15 g per tree during the entire experiment.

### 4.3. Measurement of Soil pH

Prior to soil treatment, the soil pH of the orchard was measured. Soil pH was recorded again at the fifth month after treatment to evaluate the effectiveness of the amendments. Soil pH was measured as previously described with some modifications [52]. Briefly, 20 g of air-dry soil and 100 mL of deionized water, corresponding to a soil:water ratio of 1:5 (*w:v*), were mixed, shaken for 5 min, and left to settle for 30 min. Then, soil pH was measured using a HK-3C pH meter (HUAKEYI, Beijing, China). Each soil sample was measured three times, and the average value was used for statistical analysis. The checkerboard method was used to select fifteen testing points in each grove to measure the soil pH values.

### 4.4. Detection of CLas Bacterium

The presence of CLas bacterium in the plants was confirmed using both conventional and quantitative PCR with specific primers as described previously [53]. The collected samples were identified as positive for HLB pathogen infection when the agarose gel results of conventional PCR amplification products showed clear specific bands and the cycle threshold (C_t_) values of qPCR were less than 32.0. To evaluate the impact of soil improvement on the rate of new HLB infection in the field trial, a total of three hundred randomly selected trees in each grove were divided into three groups and were assayed using PCR in November 2017. The presence or absence of CLas in new shoots of HLB-positive trees (a total of twenty HLB-positive citrus trees) was determined in November 2017. Samples of leaves (four leaves per shoot) from the asymptomatic new shoots were taken with a total of eight samples per tree. A total eighty leaf samples taken from trees in each of the groves were assayed to assess the movement of CLas bacterium to new shoots.

### 4.5. Titres Measurement of CLas Bacterium

Branches that had previously been confirmed as HLB-positive showing typical HLB symptoms were tagged at the beginning of the field trial and were used for sampling throughout the study. To estimate the dynamic changes of CLas bacterial titres over time, three sets of composite leaf samples (four leaves per branch) were collected monthly from tagged symptomatic branches from April 2017 to October 2017. Samples were washed with deionized water and dried on filter paper, and their midribs were cut into small pieces. Total DNA from the 100 mg midribs of citrus was extracted using a DNeasy Plant Mini Kit (Qiagen, Valencia, CA, USA). q-PCR analysis was performed with primers and probe to determine the titre of CLas as previously described [54]. All amplification was performed in an ABI7500 Real-Time PCR system with SYBR Fast qPCR Mix. The standard amplification protocol was 95 °C for 30 s, followed by 40 cycles at 95 °C for 5 s and 60 °C for 34 s. A standard curve was generated by performing serial dilutions of recombinant plasmid containing a fragment of 16S rRNA from CLas. The amount of test sample DNA per reaction was adjusted so that the C_t_ values were within the linear range of the standard curve. All amplification was performed in an ABI7500 Real-Time PCR system (ABI Prism 7500 (Applied Biosystems, Foster City, CA, USA) with SYBR Fast qPCR Mix (Takara, Dalian, China).

### 4.6. HLB Disease Index Assessment

HLB disease severity was visually assessed as previously described [55] from 2017 to 2019. Twenty HLB-positive trees were measured per grove. Briefly, each tree was divided into upper and lower hemisphere and each hemisphere subdivided into four equal quadrants resulting in eight sections. Each section was individually scored on a 0 to 5 scale that was indicative of the proportion of limbs expressing HLB symptoms within each section (0 = no limbs; 1 = 1 to 20% limbs; 2 = 20% to 40% limbs; 3 = 40% to 60% limbs; 4 = 60 to 80% limbs; 5 = 80 to 100% limbs). Then, the HLB disease index was evaluated and given an overall disease severity rating from 0 to 40 score for each tree (level 1 = 0 to 8 score; level 2 = 9 to 16 score; level 3 = 17 to 24 score; level 4 = 25 to 32 score; level 5 = 33 to 40 score).

### 4.7. Chlorophyll Determination

The relative chlorophyll content of the citrus plants was measured at the fifth month after transplanting using a SPAD502-puls handheld chlorophyll meter according to the instructions (KONICAMINOLTA, Tokyo, Japan). Briefly, the middle part of the citrus leaves was inserted into the detector, and the average SPAD value of the measured leaves of each plant was recorded to evaluate the relative chlorophyll content. In the field experiment, the experiments were independently repeated four times with five HLB-positive trees per replicate, and twenty leaves of the same size and maturity were collected from each tree around the canopy at the same height after five months of soil treatment. For the greenhouse experiment, the experiments were independently repeated four times with twenty HLB-positive seedlings per replicate, and four leaves were selected from each seedling.

### 4.8. Fruit Yield, Fruit Diameter and Fruit Quality

To determine fruit characteristics of HLB-infected citrus trees cultivated in varying soil conditions during the harvest season, fruit diameter and weight were recorded for each tree (a total of 20 HLB-positive trees per grove) in November 2017. Meanwhile, the pre-harvest fruit drop rate was calculated as previously described [56]. Fruit drop was counted manually, which included decomposing fruit and dry mummies under the tree canopy. After this, the amount of fruit drop was divided by the total number of fruits of each tree and multiplied by 100. To evaluate the juice quality, thirty HLB-infected fruits randomly selected from different trees in the FCK and FT were mixed as one biological replicate and then juiced by hand. The experiment was performed with three independent biological repetitions. The soluble solids (SS) content of the juice was measured with a digital PR-101 α refractometer (Atago Co, Tokyo, Japan). Briefly, the instrument was calibrated with distilled water and then 50 µL of the juice was pipetted onto the refractometer. Then, the monosaccharide (sucrose, glucose, maltose and fructose) content in the juice samples was analysed with a high performance liquid chromatography (HPLC) system following a previously described method [57]. The juice supernatant was collected using centrifugation (Eppendorf microfuge, Westbury, NY, USA) at 10,000× *g* for 30 min at 4 °C. 10 mL volume of the supernatant was added to a C-18 Sep-Pak (Waters, Milford, MA, USA), and HPLC analysis was performed on an Agilent 1200 series G1362A refractive index detector with a ZORBAX Carbohydrate Analysis columns. The analysis column (4.6 mm × 150 mm, 5 µm) (Agilent Technologies, Santa Clara, CA, USA) was operated at 30 °C at a flow rate of 0.8 mL × min^−1^. The analytes were eluted from the column with water and acetonitrile (ACN) at a 25:75 ratio. The sugar quantification was based on the external standard method (EZChrom Elite software, version 3.3.2. SP2, Santa Clara, CA, USA) with standards for sucrose, glucose, maltose and fructose. The final results were expressed in g × 100 mL^−1^.

### 4.9. Electron Microscopy Observation of Citrus Vascular Bundles

In the greenhouse trial, the GCK and GT citrus leaves with the same maturity were collected after five months of soil treatment. In brief, leaf veins were dissected and placed in a 3% glutaraldehyde fixative (pH = 7.2) for 4 h and rinsed 5 times in 0.1 M phosphate buffer and 3 times in 1% citrate buffer. Next, the samples were slowly dehydrated in a graduated ethanol series to 100% ethanol and dried under a vacuum. The surface of each sample was sprayed with a 5–10-nm-thick conductive metal coating using an EIKO IB-5 ion sputtering apparatus (EIKO, Tokyo, Japan). After the sample was progressively infiltrated with epoxy resin and polymerized at 60 °C. Finally, the morphology of the vascular bundle was observed using scanning electron microscopy (SEM) with a Hitachi SU-8010 instrument (HITACHI, Tokyo, Japan) at a beam current of 80 µA and an acceleration power of 20 kW.

### 4.10. Measurement of Root Activity Using TTC Method

For the measurement of root metabolic activity, the lateral root samples were collected at fifth months after soil treatment in the field and greenhouse trials. All experiments were independently repeated four times with five HLB-positive plants per replicate. Root metabolism activity (dehydrogenase activity) was determined spectrophotometrically at the wavelength of 520 nm based on 2,3,5-triphenyltetrazolium chloride (Coolaber, Beijing, China) as previously described [58]. Root samples (200 mg) were placed in 10 mL solution of 0.4% (*w/v*) TTC and 0.1 M sodium phosphate buffer (pH = 6.8) and vacuum-infiltrated for 15 min. After 1 h incubation at 37 °C, 0.5 mL H_2_SO_4_ (2 M) was added to terminate the reaction. The samples were then extracted in 95% (*v/v*) ethanol followed by incubation in a water bath at 90 °C for 15 min and constant volume to 5 mL. Finally, the absorbance was recorded as a quantification of root activity.

### 4.11. Determining the Endogenous Levels of SALICYLIC acid (SA)

For the salicylic acid (SA) content assay, each citrus leaf sample and root sample was collected from HLB-positive plants in the GCK and GT, respectively. Then the samples were rapidly frozen in liquid nitrogen and stored at −80 °C for later use. For each sample, 100 mg of frozen tissue power was extracted and tested for free SA as described previously [59]. The experiment was performed in three biological repetitions. In brief, the tissue was ground into powder and homogenized in 1 mL of methanol-H_2_O-acetic acid (80:19:1). After extraction overnight at 4 °C and centrifugation, the supernatant was re-extracted, and 1 mL chloroform was added and centrifugated. Then, the organic phase containing the free SA was dried in a speed vacuum with heat (~ 40 °C). The residue was resuspended in 0.5 mL of methanol, filtered and analysed using ultra performance liquid chromatography (UPLC). UPLC was performed on an ACQUITY UPLC BEHC18 column (50 mm × 2.1 mm, 1.7 μm) run at 40 °C at a flow rate of 0.4 mL × min^−1^. The analytes were eluted from the column with a mixed solvent of water with 0.1% acetic acid (solvent A) and methanol with 0.1% acetic acid (solvent B) using a linear gradient mode. The ratio of A and B was 90:10 from 0 s to 3 min, and this ratio linearly changed from 90:10 to 10:90 between 3 and 4 min. The 90:10 ratio was ultimately maintained from 4 to 7 min. The authenticity of the SA from the citrus veins and roots extract was verified based on the retention times and spectral properties, which matched perfectly with those of the commercial SA standards.

### 4.12. Gene Expression Analysis

Transcript abundance of resistance related genes *CsPR1* (LOC102627194), *CsPR5* (LOC102620891), *CsNPR1* (LOC102617188), *CsALD1* (LOC102624854), *CsWRKY1* (LOC102613235) and *CsWRKY2* (LOC102620262) in citrus leaf tissue and ion transport-related genes *CsBOR1* (LOC102624725), *CsZIP1* (LOC102627464), *CsVIT1* (LOC102627805) and *CsAmtB* (LOC102619818) in lateral root tissue were analysed using qPCR in the greenhouse trial. These experiments were performed in three biological repetitions, respectively. Each biological replicate of leaf sample consisted of nine leaves combined from three different plants and each biological replicate of root sample consisted of combined root systems from three different plants. The total RNA from the midribs and lateral roots of citrus was extracted using a plant RNA kit (Qiagen, Valencia, CA). RNA integrity was verified using a 1.2% formaldehyde agarose gel, and 1 µg of total RNA was reverse transcribed with PrimeScript RT Reagent Kit (Takara, Dalian, China) according to the manufacturer’s instructions. All amplification was performed in an ABI7500 Real-Time PCR system with SYBR Fast qPCR Mix. The standard amplification protocol was 95 °C for 5 min. Then, the following cycle was repeated 40 times: 95 °C for 15 s, 60 °C for 15 s, and 72 °C for 15 s. The housekeeping gene of citrus encoding glyceraldehyde-3-phosphate dehydrogenase (*GAPDH*) was used as an internal reference to normalize the amount of RNA in the different reactions. The relative mRNA quantities were calculated using the ^ΔΔ^Ct method [60]. Gene specific primers used for q-PCR are shown in Appendix A.

### 4.13. Protein Extraction

Six citrus leaf midrib samples were selected (three biological replicate samples from the GCK and the others from the GT) in October 2017. The sample was ground in a mortar in the presence of liquid nitrogen, and 10 mL of lysis buffer (8 M urea, 2% SDS, 1× Protease Inhibitor Cocktail (Roche Ltd. Basel, Switzerland)) was added to each sample power, mixed using a vortex, then sonicated on ice and centrifugated at 13,000× *g* for 15 min at 4 °C. The supernatant was filtered using a 0.22 μm filter and precipitated with ice-cold acetone at −20 °C overnight. The precipitations were cleaned with 50% ethanol and 50% acetone three times. 200 μg proteins was diluted by buffer (100 mM Tris, pH = 8.0, 8 M urea) to 100 μL, and then the solution was added to 11 μL DTT (1 M) and incubated at 37 °C for 1 h. The treated samples were put in a 10 kDa ultrafiltration tube (Millipore, Billerica, MA, USA) and centrifuged at 12,000× *g* for 10 min. Then, 100 μL 55 mM iodoacetamide (IAA) was added to block reduce cysteine residues and it was incubated for 20 min in darkness at room temperature.

### 4.14. Proteomic TMT and Data Analysis

For the TMT experiments, the procedure was based on the manufacturer’s protocols (Abmart Ltd., Shanghai, China). The MS/MS spectra were analysed using the MASCOT engine (Matrix Science, London, UK; version 2.5.1). This search engine was set up to search the UniProt *Citrus sinensis* database assuming digestion with trypsin. The MASCOT search was conducted with a fragment ion mass tolerance of 0.020 Da and a parent ion tolerance of 5.0 ppm. Scaffold Q+ (version Scaffold_4.6.2, Proteome Software Inc., Portland, OR, USA) was used to quantitate TMT Label Based Quantitation peptide and protein identifications. Normalization was performed iteratively (across samples and spectra) on intensities, as described in Statistical Analysis of Relative Labeled Mass Spectrometry Data from Complex Samples Using ANOVA. Medians were used for averaging. Spectra data were log-transformed, pruned of those matched to multiple proteins and weighted using an adaptive intensity weighting algorithm. Differentially expressed proteins were determined by applying a Mann-Whitney test with unadjusted significance level *p* < 0.05 corrected by Benjamini-Hochberg.

### 4.15. Statistical Data Analysis

The biological data reported in this study was analysed as follows using SPSS 20 software (SPSS Inc., Chicago, IL, USA). To determine whether there were significant differences between treatment and control groups, a Student’s *t*-test was performed. Differences were considered significant at * *p* < 0.05 and ** *p* < 0.01.

## 5. Conclusions

The critical challenge of HLB disease management in southern China, where citrus orchards have severe soil acidification and HLB disease prevalence, is to maintain the sustainable economic production of CLas-infected citrus. In addition to traditional methods for the control of HLB disease, we should pay more attention to the role of environmental factors, which are one of the important components of the HLB disease triangle in mitigating the destructive impact on citrus production. Liming is a manageable and effective approach to ameliorate acidic soil, which is conducive to preserve root activity and strengthen immune defence against CLas bacterium. In this study, we found that the incidence of HLB was reduced and the progression was postponed after acid soil improvement in field experiments, resulting in increased yield and fruit quality of citrus in liming-amended soils. However, the generalisability of these results is subject to certain limitations because the intricate nature of the HLB pathosystem makes it difficult to completely control the disease through single aspect management. Therefore, we hope that the point of view presented herein will motivate more investigation to establish an integrated control strategy involving soil environment optimization against CLas infection.

## Figures and Tables

**Figure 1 ijms-21-03614-f001:**
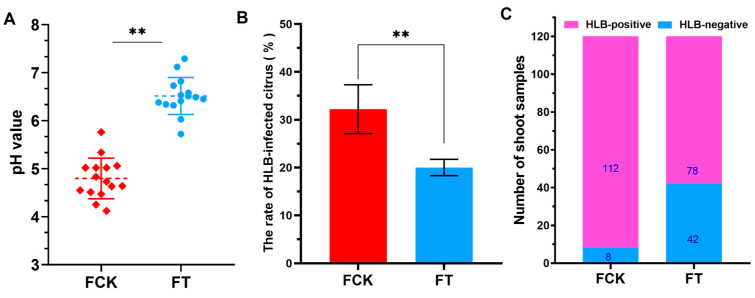
Effect of acid soil improvement on the spread and movement of Huanglongbing (HLB) disease in the field trial. (**A**) Soil pH value in citrus orchards. The blue dots represent the soil pH before treatment, and the red dots represent the soil pH at the fifth month after treatment. FCK, control grove in the field trial; FT, treatment grove in field trial. (**B**) The new infections rate of HLB disease in FCK and FT at the fifth month after soil treatment. Data represent the mean ± SD of three independent biological replicates. Asterisks indicate a significant difference according to the Student’s *t*-test: ** *p* < 0.01. (**C**) Detection of CLas bacterium in the leaves of new asymptomatic shoots of HLB-infected citrus trees in FCK and FT.

**Figure 2 ijms-21-03614-f002:**
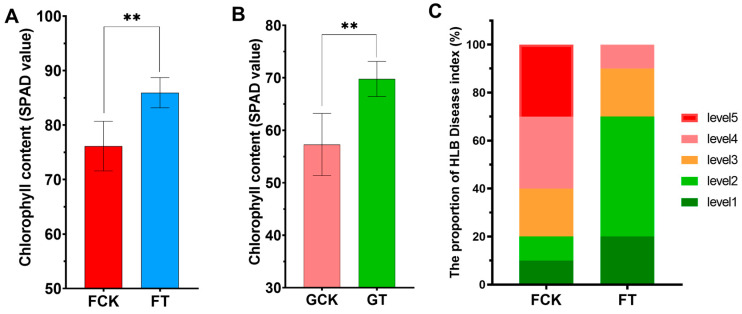
Effect of acid soil improvement on the content of chlorophyll in infected leaves and the progression of Huanglongbing (HLB) disease. The relative chlorophyll content (SPAD value) of HLB-infected citrus leaves at the fifth month after soil treatment in the field trial (**A**) and greenhouse trial (**B**). Data represent the mean ± SD of four independent biological replicates. (**C**) HLB disease index was assessed by measuring the proportion of limbs expressing HLB symptoms of each tree at the fifth month after soil treatment (*n* = 20). FCK, control grove in field trial; FT, treatment grove in field trial. GCK, control group in greenhouse trial, GT, treatment group in greenhouse trial. Asterisks indicate a significant difference according to the Student’s *t*-test: ** *p* < 0.01.

**Figure 3 ijms-21-03614-f003:**
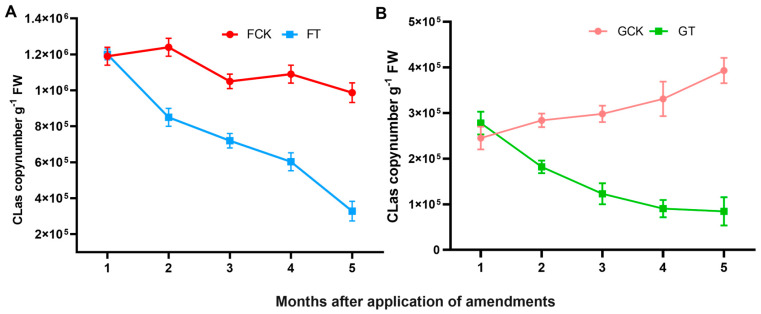
Temporal variation of the titre of ’*Candidatus* Liberibacter asiaticus’ (CLas) in leaf midrib samples under different Soil acidity conditions. (**A**) In the field and (**B**) greenhouse trials, the CLas titre in infected leaves were measured at each time point using q-PCR. Data represent the means (three biologically independent experiments for quantification of CLas) ± SD. FCK, control grove in field trial; FT, treatment grove in field trial; GCK, control group in greenhouse trial; GT, treatment group in greenhouse trial; FW, fresh weight.

**Figure 4 ijms-21-03614-f004:**
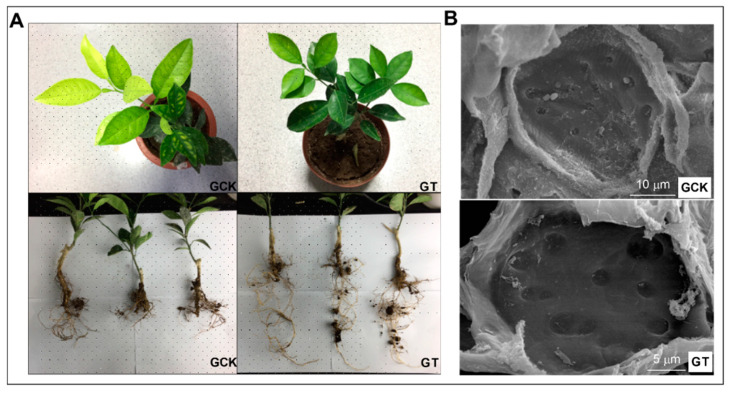
Variation of physiology and morphology of HLB-infected citrus seedlings under different soil acidity conditions in the greenhouse trial. Physiological observations of citrus roots and leaves (**A**) and scanning electron microscope (SEM) observations of citrus sieve tubes (**B**) of HLB-infected citrus seedlings in GCK (soil pH 4.5) and GT (soil pH 6.5). All samples were collected at the fifth month after being repotted. GCK, control group in greenhouse trial, GT, treatment group in greenhouse trial.

**Figure 5 ijms-21-03614-f005:**
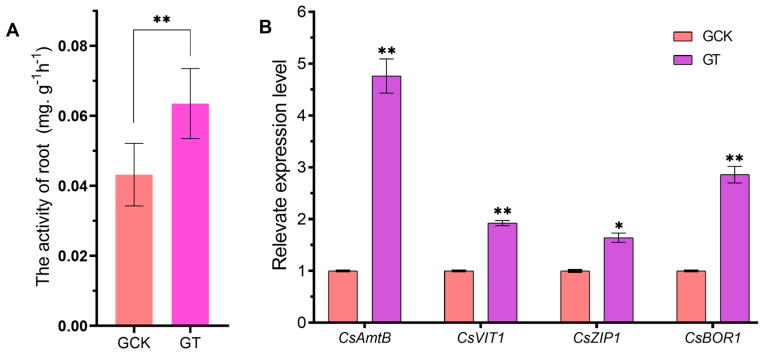
Effect of acid soil improvement on root activity and transcriptional expression of ion transporter-related genes of HLB-infected citrus seedlings in the greenhouse trial. (**A**) The root activity of HLB-infected citrus seedling samples in the GCK and GT was measured using the TTC (2,3,5-triphenyltetrazolium chloride) method at the fifth month after repotting. Data represent the mean ± SD of four independent biological replicates. (**B**) Relative expression levels of ion transporter genes in roots of HLB-infected citrus in the GCK and GT at the fifth month after repotting. Data represent the means (three biologically independent experiments for gene expression) ± SD. GCK, control group in greenhouse trial; GT, treatment group in greenhouse trial. Asterisks indicate a significant difference according to the Student’s *t*-test: * *p* < 0.05, ** *p* < 0.01.

**Figure 6 ijms-21-03614-f006:**
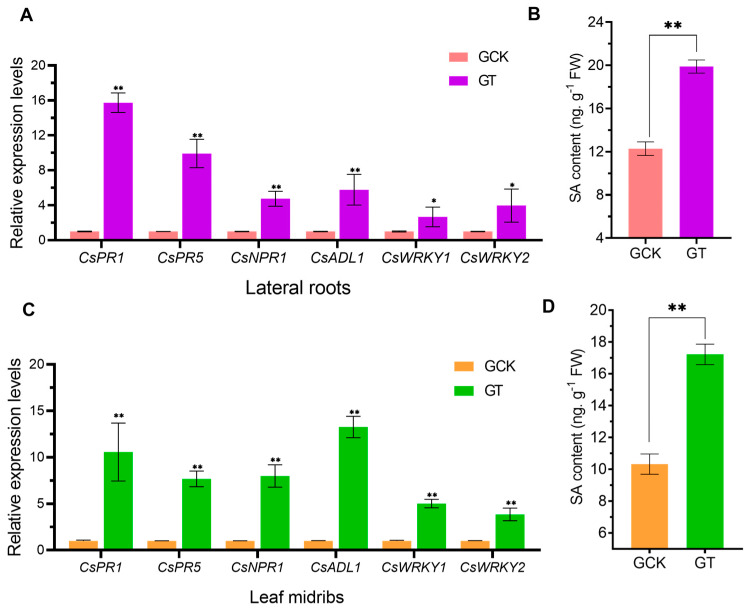
Effect of acid soil improvement on transcriptional expression of defence-related genes and SA content in HLB-infected citrus seedlings in the greenhouse trial. Relative expression levels of six plant defence-related genes in lateral roots (**A**) and leaf midribs (**C**) of HLB-infected citrus seedlings in GCK and GT at the fifth month after repotting. Data represent the means (three biologically independent experiments for gene expression) ± SD. Salicylic acid (SA) content determined using high-performance liquid chromatography (HPLC) in roots (**B**) and midribs (**D**) of HLB-infected citrus seedlings in GCK and GT at the fifth month after repotting. Data represent the mean ± SD of three independent biological replicates. GCK, control group in greenhouse trial; GT, treatment group in greenhouse trial; FW, fresh weight. Asterisks indicate a significant difference according to the Student’s *t*-test: * *p* < 0.05, ** *p* < 0.01.

**Table 1 ijms-21-03614-t001:** Fruit yield and quality of HLB-infected citrus cultivated under two soil conditions.

	FT	FCK
**Production index**		
Yield (kg × tree^−1^)	57.6 ± 2.40 *	52 ± 3.00
Fruit diameter (cm)	7.86 ± 0.19 *	7.25 ± 0.31
Fruit drop rate (%)	22.25 ± 1.90 **	47.43 ± 3.40
**Fruit quality index**		
Soluble solids (%)	14.5 ± 0.62 *	11.3 ± 1.21
Vitamin C (mg × 100 g^−1^)	38.4 ± 0.92	39.8 ± 2.32
Glucose (g × 100 g^−1^)	2.51±0.21 *	1.51 ± 0.10
Sucrose (g × 100 g^−1^)	4.40 ± 0.11 **	0.98 ± 0.08
Fructose (g × 100 g^−1^)	2.40 ± 0.10 **	1.50 ± 0.12
Maltose (g × 100 g^−1^)	0.43 ± 0.13	ND

Note: Data of production index shown are mean ± SD of twenty replications. Data of fruit quality index shown are the mean ± SD of three independent biological replicates. ND, not determined; FCK, control grove in field trial; FT, treatment grove in field trial. Asterisks indicate a significant difference according to the Student’s *t*-test: * *p* < 0.05, ** *p* < 0.01.

**Table 2 ijms-21-03614-t002:** Up-regulated proteins involved in defence responses and secondary metabolic pathways in midribs of HLB-infected citrus seedlings cultivated in amended soil conditions. The detailed information of all different expression proteins (DEPs) is listed in Appendix A.

UniProtKB ID	Fold Change	*p*-Value	Description	Species
**Defence/stress responses**				
A0A067H9X4	2.58	0.0039	Thaumatin-like protein (PR5)	*Dorcoceras hygrometricum*
A0A067DC18	1.6	0.0001	Pathogenesis-related protein 1	*Arabidopsis thaliana*
A0A067FFK4	1.51	0.0001	Lipoxygenase	*Citrus sinensis*
A0A067DHQ0	1.53	0.0001	Like serine/threonine-protein kinase	*Citrus sinensis*
**Secondary metabolism**				
A0A067FUD3	2.53	0.0007	Salicylate carboxymethyltransferase	*Handroanthus impetiginosus*
A0A067DZ96	1.51	0.0007	SABATH methyltransferase 3	*Lonicera japonica var. chinensis*
A0A067E1I5	1.84	0.0003	Caffeic acid 3-O-methyltransferase	*Zea mays*
A0A067DNI8	1.51	0.009	Caffeic acid 3-O-methyltransferase	*Zea mays*
A0A067DIU3	1.55	0.0001	Orcinol O-methyltransferase	*Medicago truncatula*
A0A067DU63	1.82	0.0011	Terpene synthase	*Citrus sinensis*
A0A067F9Y6	1.72	0.0039	Dirigent protein	*Citrus sinensis*
A0A067FU00	1.67	0.0001	Dirigent protein	*Citrus sinensis*
A0A067F6J2	1.62	0.0001	Dirigent protein	*Citrus sinensis*
A0A067D7I5	1.71	0.0003	Serine-type carboxypeptidase	*Citrus sinensis*
A0A067EBY4	1.54	0.009	Serine-type carboxypeptidase	*Citrus sinensis*
A0A067D666	1.51	0.0001	Serine-type carboxypeptidase	*Citrus sinensis*
A0A067GBM2	1.51	0.0001	Putative UDP-glucosyltransferase	*Citrus paradisi*
A0A067FCS1	1.51	0.0001	Beta_HSD domain-containing protein	*Citrus sinensis*

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
