# Peer review of "Acid Soil Improvement Enhances Disease Tolerance in Citrus Infected by Candidatus Liberibacter asiaticus"

_ijms, 2020, doi:10.3390/ijms21103614_

Round 1
Reviewer 1 Report
This manuscript is valuable as it firstly demonstrates that ameliorating soil acidity improves plant health and symptom abatement, then investigates associations with a range of indicators of the physiological health and activation of defence pathways in plants grown on amended soils to attempt to explain these effects. No clear mechanism linking soil pH and activation of defences emerges, and it could involve either or both direct or indirect mechanisms. An example of an indirect mechanism might be that reducing soil acidity improves the diversity and activity of beneficial soil microbes, as suggested, as well as colonisation of roots by endophytes and mycorrhizae. This would be worth further investigation.
The study only compares the association between liming, symptoms and defence activation in infected plants. It should have been possible from the design of the field trial to include measurements of defence activation and other physiological parameters in uninfected plants. This would have enabled the authors to distinguish changes in host physiology due to soil pH alone from changes due the effect of soil pH on infection and symptom development.
One small correction is required in the caption to Figure 3 that refers to "C" and "D". I assume this should read "A" and "B"?
A major outcome of this work is the observation that liming can improve yields and quality, and could thus have significant effects on the incomes of citrus farmers. Liming is relatively inexpensive and environmentally benign.
Author Response
Dear reviewer,
We are really glad to hear from your positive comments about our work. All authors appreciate your efforts on this paper. We have made modifications based on your suggestions.
Point 1: The study only compares the association between liming, symptoms and defence activation in infected plants. It should have been possible from the design of the field trial to include measurements of defence activation and other physiological parameters in uninfected plants. This would have enabled the authors to distinguish changes in host physiology due to soil pH alone from changes due the effect of soil pH on infection and symptom development.
Response 1: Thanks for your professional suggestion. We believe that it is very important to study the changes of healthy citrus under different soil pH conditions, especially for exploring the mechanism of soil pH normalization to improve disease resistance. We deeply hope to strengthen the the research of this part and improve the experimental design according to your suggestions in the future work.
Point 2: One small correction is required in the caption to Figure 3 that refers to "C" and "D". I assume this should read "A" and "B"?
Response 2: The "C" and "D" in Figure 3 has been replaced with "A" and "B", respectively.
Reviewer 2 Report
The results of these experiments demonstrate that normalizing soil pH using lime hinders the progression and severity of HLB disease. While the evidence is clear, the mechanism of action is less clear. The title and conclusions state that soil pH normalization enhances disease tolerance, but it is not clear to me if pH normalization simply makes healthier citrus that allows for better disease resistance. It is a small distinction yet I think makes a difference. Meaning, if they take healthy citrus and expose to the two different soil pH levels, would they also observe significant differences in the measured gene expression and fruit size? I am also a bit curious about the results in Figure 3 and how it fits with the other results they observed (of increasing disease) and wonder if the authors care to discuss. There are also mistakes using the word disease and pathogen in multiple areas of the manuscript. For example, the title CLas and HLB are reversed – disease tolerance would be to the disease, HLB, and citrus are infected with the pathogen CLas (not the disease).
There are a number of grammatical mistakes in the document. Plenty of sentences are awkward or unclear, and in a few places they mix up their control and treated when presenting the results. One example of the latter is lines 99-102. Another example is L120-122, and the column labeling of Table 1 is backwards I think. Examples of awkward or incorrect sentences are L13 remove “disease” after HLB, L22 “disoders” should be “disorders”, L44 “maintains” to “maintain” though still awkward, L59-60 (in recent 20 years) change to “in the past 20 years”, L69-70, L75 “they” instead of “them”; “to prevent” instead of “for preventing”, L79 “orchards”, L124 is awkward, L154 “C and D” should be “A, B” respectively, L185 “trail” should be “trial”, L186 please define TTC, Figure 4 is labeled FCK and FT in the figures and should be GCK and GT, L223 “folds” should be “fold”, many lines (e.g. 24, 247, 248, 255, Table 2, 356) “defence” should be “defense”, L252 “The most of…” should be “Most…” and remove mostly on the next line, L272 remove “merely”, L283-284 is awkward because HLB causal agents do not live in the soil, L439 “evaluated” should be “evaluate”, L450 “monthly collected” change to “collected monthly”.
Author Response
Response to Reviewer 2 Comments
Point 1: The title and conclusions state that soil pH normalization enhances disease tolerance, but it is not clear to me if pH normalization simply makes healthier citrus that allows for better disease resistance. It is a small distinction yet I think makes a difference. Meaning, if they take healthy citrus and expose to the two different soil pH levels, would they also observe significant differences in the measured gene expression and fruit size?
Response 1: The acidic soil is not suitable for the growth of HLB-free citrus, if soil pH was normalized, we think it is likely to enhance yield (fruit size) of citrus due to growing environment is improved. Previous researchers found that, in most cases, acid soils reduce crop yields worldwide. Actually, any environmental factor, in particular pH, that affects the acquisition of water or mineral nutrients by the root system will affect crop yields[1]. In this study, we indeed did not survey the changes in the expression of resistance-related genes when healthy citrus was exposed to the different soil pH levels. The reason is that: in our study, the HLB-infected citrus trees were given priority as research object because HLB is widespread in southern China. Farmers are more concerned about the established, HLB infected citrus trees and how to reduce the severity of HLB and maintain the economic benefits of orchards. Meanwhile, it is very important to study the changes of healthy citrus under different soil pH conditions, especially for exploring the mechanism of soil pH normalization to improve disease resistance in HLB-free citrus. The related works will be further studied in the next step.
Point 2: I am also a bit curious about the results in Figure 3 and how it fits with the other results they observed (of increasing disease) and wonder if the authors care to discuss.
Response 2: Whether the question you raised is the correlation between the CLas titre in leaves of FCK (Figure 3A) and the disease severity index in field trial. The titre of CLas is indeed closely associated with the degree of yellowing in infected leaves with typical symptoms, it can represent the propagation and accumulation status of CLas in infected citrus trees at the microcosmic level. Although our results show that the CLas titre fluctuates over time, the movement and colonization of the CLas in infected branches to the new leaves of other parts of the citrus can still proceed normally.
HLB disease severity of the whole canopy was visually assessed according to the proportion of the limbs expressing yellow shoot (the leaf yellowing degree is not used as a reference standard for evaluating the disease index in this study) at the tree level. We consider these two indicators as independent parameters for evaluating the development of HLB disease progression.
Point 3: There are also mistakes using the word disease and pathogen in multiple areas of the manuscript. For example, the title CLas and HLB are reversed-disease tolerance would be to the disease, HLB, and citrus are infected with the pathogen CLas (not the disease).
Response 3: The title was changed to “Acid soil improvement enhances disease tolerance in citrus infected by Candidatus Liberibacter asiaticus”.
Point 4: There are a number of grammatical mistakes in the document. Plenty of sentences are awkward or unclear, and in a few places they mix up their control and treated when presenting the results.
Response 4: We appreciate your careful reading and professional suggestions. We feel ashamed to have these spelling and format mistakes and try to check all.
- One example of the latter is lines 99-102, L120-122.
The “FCK” and “FT” were replaced with “FT” and “FCK”, respectively.
- Another example is the column labeling of Table 1 is backwards I think.
The error of Table 1 has been corrected.
- Examples of awkward or incorrect sentences are L13 remove “disease” after HLB,
The “disease” was removed from this sentences.
- L22 “disoders” should be “disorders”,
The “disoders” was changed to “disorders”.
- L44 “maintains” to “maintain” though still awkward,
For better understanding, the sentence was modified to “Therefore, it is urgent to develop a sustainable solution to suppress the spread of HLB disease and enable the continued economic production of CLas-infected citrus trees”.
- L59-60 (in recent 20 years) change to “in the past 20 years”.
The “in recent 20 years” was changed to “in the past 20 years”.
- L69-70, L75 “they” instead of “them”; “to prevent” instead of “for preventing”,
The “them” and “for preventing” have been replaced with “them” and “to prevent”, respectively.
- L79 “orchards”,
The “orchard” was changed to “orchards”.
- L124 is awkward,
This sentence was modified to “leaves in GCK showed obvious symptoms of spotted-yellowing at the fifth month after soil treatment, whereas typical symptoms did not appear on the leaves of GT (Figure 4A)”.
- L154 “C and D” should be “A, B” respectively,
The “C and D” was corrected to “A and B”, respectively.
- L185 “trail” should be “trial”,
The “trail” was changed to “trial”.
- L186 please define TTC,
The TTC has been defined as “TTC (2,3,5-triphenyltetrazolium chloride)”.
- Figure 4 is labeled FCK and FT in the figures and should be GCK and GT,
The labeled “FCK and FT” was corrected to “GCK and GT”.
- L223 “folds” should be “fold”, many lines (e.g. 24, 247, 248, 255, Table 2, 356) “defence” should be “defense”,
The “folds” and “defence” were replaced by “fold” and “defense”, respectively.
- L252 “The most of…” should be “Most…” and remove mostly on the next line,
The “The most of…” was changed to “Most…” and “mostly” was removed.
- L272 remove “merely”,
The “merely” was removed.
- L283-284 is awkward because HLB causal agents do not live in the soil,
For better understanding, we have modified the sentence into “According to these data, we can infer that soil acidification promotes the occurrence of HLB disease”.
- L439 “evaluated” should be “evaluate”,
The “evaluated” was changed to “evaluate”.
- L450 “monthly collected” change to “collected monthly”.
The “monthly collected” was changed to “collected monthly”.
[1]. Holland, J.E.; Bennett, A.E.; Newton, A.C.; White, P.J.; Mckenzie, B.M.; George, T.S.; Pakeman, R.J.; Bailey, J.S.; Fornara, D.A.; Hayes, R.C. Science of the Total Environment Liming impacts on soils , crops and biodiversity in the UK : A review. Sci. Total Environ. 2018, 610–611, 316–332.